# Paraoxonase-1 Serum Concentration and *PON1* Gene Polymorphisms: Relationship with Non-Alcoholic Fatty Liver Disease

**DOI:** 10.3390/jcm8122200

**Published:** 2019-12-13

**Authors:** Mircea Vasile Milaciu, Ștefan Cristian Vesa, Ioana Corina Bocșan, Lorena Ciumărnean, Dorel Sâmpelean, Vasile Negrean, Raluca Maria Pop, Daniela Maria Matei, Sergiu Pașca, Andreea Liana Răchișan, Anca Dana Buzoianu, Monica Acalovschi

**Affiliations:** 1Department 5—Internal Medicine, 4th Medical Clinic, Faculty of Medicine, “Iuliu Haţieganu” University of Medicine and Pharmacy, 400015 Cluj-Napoca, Romania; mircea_milaciu@yahoo.com (M.V.M.); dorel.sampelean@gmail.com (D.S.); Vasile.Negrean@umfcluj.ro (V.N.); 2Department 2—Functional Sciences, Discipline of Pharmacology, Toxicology and Clinical Pharmacology, Faculty of Medicine, “Iuliu Haţieganu” University of Medicine and Pharmacy, 400337 Cluj-Napoca, Romania; stefanvesa@gmail.com (Ș.C.V.); corinabocsan@yahoo.com (I.C.B.); raluca_parlog@yahoo.com (R.M.P.); abuzoianu@umfcluj.ro (A.D.B.); 3Department 5—Internal Medicine, 3rd Medical Clinic, Faculty of Medicine, “Iuliu Haţieganu” University of Medicine and Pharmacy, 400162 Cluj-Napoca, Romania; dmatei68@gmail.com; 4Faculty of Medicine, “Iuliu Haţieganu” University of Medicine and Pharmacy, 400012 Cluj-Napoca, Romania; pasca.sergiu123@gmail.com; 5Department of Pediatrics, “Iuliu Haţieganu” University of Medicine and Pharmacy, 400177 Cluj-Napoca, Romania; andreea_rachisan@yahoo.com; 6Doctoral School, “Iuliu Haţieganu” University of Medicine and Pharmacy, 400012 Cluj-Napoca, Romania; monacal@umfcluj.ro

**Keywords:** non-alcoholic steatohepatitis, paraoxonase-1, gene polymorphisms

## Abstract

Background: Non-alcoholic fatty liver disease (NAFLD) is an important cause of chronic liver diseases around the world. Paraoxonase-1 (PON1) is an enzyme produced by the liver with an important antioxidant role. The aim of this study was to evaluate PON1 serum concentration and *PON1* gene polymorphisms in patients with NAFLD. Materials and methods: We studied a group of 81 patients with NAFLD with persistently elevated aminotransferases and a control group of 81 patients without liver diseases. We collected clinical information and performed routine blood tests. We also measured the serum concentration of PON1 and evaluated the *PON1* gene polymorphisms L55M, Q192R, and C-108T. Results: There was a significant difference (*p* < 0.001) in serum PON1 concentrations among the two groups. The heterozygous and the mutated homozygous variants (LM + MM) of the L55M polymorphism were more frequent in the NAFLD group (*p* < 0.001). These genotypes were found in a multivariate binary logistic regression to be independently linked to NAFLD (Odds ratio = 3.4; *p* = 0.04). In a multivariate linear regression model, the presence of NAFLD was associated with low PON1 concentration (*p* < 0.001). Conclusions: PON1 serum concentrations were diminished in patients with NAFLD, and the presence of NAFLD was linked with low PON1 concentration. The LM + MM genotypes of the *PON1* L55M polymorphism were an independent predictor for NAFLD with persistently elevated aminotransferases.

## 1. Introduction

Over the last decades, non-alcoholic fatty liver disease (NAFLD) and non-alcoholic steatohepatitis (NASH) have gained increasing ground in clinical research. This can be explained partially due to the “epidemic” of obesity and metabolic syndrome and due to the potential of its evolution from simple liver steatosis to NASH, advanced fibrosis, cirrhosis, and hepatocellular carcinoma [1,2,3]. These diseases have yet to be assigned a specific treatment [4], and their gold-standard for diagnosis (i.e., liver biopsy) has begun to become unsuitable and unrealistic in clinical practice, given the large number of patients suffering from NAFLD/NASH [5]. Recently, a group of renowned hepatologists concluded that it is of the utmost importance to develop non-invasive modalities which will be able to diagnose NASH and evaluate its fibrosis progression [6]. 

Paraoxonase-1 (PON1) is an enzyme synthesized mainly in the liver and is capable of hydrolyzing peroxides and lactones associated with lipoproteins. Its association in the blood with high-density lipoproteins (HDLs) results in an atheroprotective effect, mainly due to the fact of its ability to prevent low-density lipoprotein (LDL) oxidation. This enzyme has the ability to hydrolyze various types of substrates thus having the complex role of defending against an increase in oxidative stress [7,8,9]. Recently, it was stated that PON1 is linked to a reduction in oxidative stress and inflammation, and it has already been clearly linked to several diseases that express high levels of inflammation such as diabetes mellitus [10].

Few studies have evaluated PON1 activities and the *PON1* genotype in patients with liver disorders [11,12,13]. Even fewer studies have been conducted on patients with NAFLD or NASH [14,15], some of these evaluating pediatric populations [16]. Most of these works showed PON1 activities to be decreased in patients with NAFLD/NASH, but, in many cases, the results were inconsistent, leaving many questions rather than clarifying the impact of PON1 in this liver disease. 

The main aim of this study was to examine the *PON1* gene polymorphisms’ role as a risk factor for NAFLD with persistently elevated aminotransferases levels. Another objective was to evaluate PON1 serum concentrations in patients with NAFLD which, to the best of our knowledge, has never previously been performed. The last objective was to find the factors that influence PON1 serum concentration levels.

## 2. Materials and Methods 

The study was analytical, observational, prospective, transversal, and case-control type. It was conducted between July 2015 and July 2019 in the Clinical CF University Hospital in Cluj-Napoca, Romania. The study was approved by the Ethics Committee of the “Iuliu Hațieganu” University of Medicine and Pharmacy (No. 404/02/Jul/2015) and was conducted according to the Declaration of Helsinki. All subjects signed a consent form for study participation. We included a group of 81 patients diagnosed with NAFLD and 81 controls in whom NAFLD was excluded, who were age (±1 year) and gender matched and selected according to strict inclusion and exclusion criteria after a minimum of 6 months prior to screening. Inclusion criteria: patients diagnosed with NAFLD by ultrasonographically proven liver steatosis (as evaluated by the same experienced operator) and persistently elevated aminotransferases (on more than two different occasions, with more than six months between testing) with negative markers for viral hepatitis or other liver disorders (autoimmune hepatitis, primary biliary cirrhosis or cholangitis, hemochromatosis, Wilson’s disease, etc.). Exclusion criteria: significant alcohol consumption (≥30 g/day for men and ≥20 g/day for women as defined in the 2010 position statement of the European Association for the Study of Liver [17]), pregnancy, use of medication with potential liver toxicity, and any other disease that might influence PON1 activity (dysfunctions of the thyroid gland, autoimmune diseases, malignancies, psychiatric disorders, etc.). 

We collected general information about each patient: age, gender, body mass index (BMI), waist circumference, family history of cardiovascular diseases, and other comorbidities (i.e., hypertension, ischemic heart disease with or without angina pectoris, pre-diabetes, diabetes mellitus type 1 or 2, and metabolic syndrome). A blood sample was obtained from each patient for routine measurements (i.e., glycemia, aspartate aminotransferase (AST), alanine aminotransferase (ALT), alkaline phosphatase (ALP), gamma-glutamyl transferase (GGT), platelets count (PLT), serum bilirubin, total cholesterol, High-density lipoprotein Cholesterol (HDL-cholesterol), triglycerides, and albumin). Another blood sample was taken for DNA extraction which was performed using a specific kit (PureLink^®^ Genomic DNA Mini Kit, Invitrogen). The last sample was centrifuged and stored as serum at −20 °C (later being used for specific testing of PON1 serum concentration). 

Abdominal ultrasound was performed by an experienced physician using an Aloka Prosound Alpha 7 Premier ultrasound machine, using a convex transducer. 

Although liver biopsy remains the golden standard for the diagnosis of NASH [2], we respected what was recommended in 2014 [18], i.e., to be used only if diagnostic uncertainty due to the fact of its invasiveness; thus, our patients did not undergo a liver biopsy [18]. We were not able to establish a NASH diagnosis in our patients using only abdominal ultrasound-proven steatosis and moderately elevated serum levels of aminotransferases (on more than two different occasions during a minimum six months follow-up). However, after exclusion of any other causes of hepatitis and of significant alcohol consumption, it is likely that most of these patients had NASH at the moment of their inclusion in the study. 

The PON1 serum concentration levels were determined using Human PON1 (Paraoxonase-1) ELISA Kits which use PON1 antibody–PON1 antigen interactions and a colorimetric detection system to identify PON1 antigen in serum samples. 

We evaluated the status of the *PON1* gene polymorphisms L55M, Q192R (from the coding part of the gene), and C-108T, situated in the regulatory part of the gene, using a PCR-RFLP method as we described in detail in our previous study [19]. 

Statistical analysis was performed using MedCalc Statistical Software version 19.1 (MedCalc Software bv, Ostend, Belgium; https://www.medcalc.org; 2019). Continuous data were evaluated for normality of distribution (Shapiro–Wilk test and histograms) and expressed as the median and 25th–75th percentiles. Qualitative data were characterized by frequency and percentage. Comparisons among groups were performed using the Mann–Whitney of chi-square tests when appropriate. Correlations among variables were verified using the Spearman’s rank correlation coefficient. Chi-square tests were used to verify the Hardy–Weinberg equilibrium. Multivariate logistic regression was used in order to determine which variables were associated with the presence of NAFLD. Multivariate linear regression was used in order to determine which parameters were associated with PON1 serum concentration. Due to the fact that PON1 values followed a non-normal distribution, we used a logarithmic transformation. A *p*-value < 0.05 was considered statistically significant.

## 3. Results

The clinical data recorded for each subject from our study groups are shown in Table 1.

Each group consisted of 38 women and 43 men. Diabetes and pre-diabetes were more frequent in the NAFLD group (55.6% of cases). Also, hypertension and metabolic syndrome were more frequent in the NAFLD group (*p* < 0.001). There was a statistically significant difference (*p* < 0.001) in the aminotransferase serum levels, alkaline phosphatase, gamma-glutamyl transpeptidase, glycemia, HDL cholesterol, triglycerides, and serum albumin levels among the two groups.

The measured serum values of PON1 concentration and the variations in the three polymorphisms of the *PON1* gene are recorded in Table 2. The PON1 concentration was statistically significantly reduced in the NAFLD group (*p* < 0.001).

The L55M polymorphism had the greatest variability among the groups, with the heterozygous variant LM present in 58% of the patients in the NAFLD group and only in 33.3% of the patients in the control group. The wild genotype LL was present in 27.2% of patients in the NAFLD group and in 54.3% in the non-NAFLD subjects. We found a statistically significant difference among the two groups (*p* = 0.002). Also, taking into consideration the presence on the M allele, the percentage of subjects having either the heterozygous (LM) or mutated homozygous (MM) genotype (LM + MM) was 72.8% in the NAFLD group and only 45.7% in the other cohort (*p* < 0.001). The frequency of the M allele was 0.438 in the NALFD group and 0.2895 in the control group (*p* = 0.05). We did not observe a statistically significant difference between the distribution in the two groups of the Q192R and C-108T polymorphisms (*p* = 0.807 and 0.707, respectively). We did not observe a difference among groups regarding the frequency of the R allele (*p* = 0.731) or the T allele (*p* = 0.451).

A multivariate binary logistic regression (stepwise selection method–forward selection conditional) was used to create several models in order to determine which variables were independently associated with the risk of NAFLD (Table 3). The most relevant model included the variables BMI, family history of cardiovascular diseases, hypertension, diabetes mellitus, ischemic heart disease, and L55M polymorphism. We calculated a Cox–Snell *R*^2^ of 0.588. The variables that were independently linked to NAFLD were BMI (OR = 2.1; *p* < 0.001), family history of cardiovascular diseases (OR = 4.3; *p* = 0.02), and the LM + MM genotypes of the L55M polymorphism (OR = 3.4; *p* = 0.04).

The influence of different variables on PON1 concentration is shown in Table 4. A statistically significant influence on PON1 concentration was observed for the family history of cardiovascular disease (*p* < 0.001), metabolic syndrome (*p* < 0.001), hypertension (*p* = 0.004), pre-diabetes (*p* = 0.004), and L55M polymorphism (*p* = 0.015), most particularly for the LM + MM variants (*p* = 0.004). Also, using the Spearman’s rho, we correlated the PON1 concentration with the other variables (previously presented in Table 1 using the median and the 25th/75th percentiles). Thus, we observed a statistically significant correlation between PON1 concentration and BMI (*p* < 0.001), waist circumference (*p* < 0.001), albumin (*p* = 0.006), HDL cholesterol (*p* = 0.002), triglycerides (*p* < 0.001), AST (*p* < 0.001), ALT (*p* < 0.001), alkaline phosphatase (*p* < 0.001), and gamma-glutamyl transpeptidase (*p* = 0.001).

A multivariate linear regression was used in order to find which variables influence PON1 concentrations. Due to the fact that the PON1 concentrations followed a non-normal distribution, we performed a logarithmic transformation. We found an adjusted R^2^ coefficient of 0.420. In this regression model, NAFLD was statistically significantly associated to PON1 concentration (*p* < 0.001) (Table 5).

## 4. Discussion

Non-alcoholic fatty liver disease and especially NASH still have many unsolved questions regarding pathogenesis, minimally invasive options for diagnosis, and treatment. Some studies have even suggested the possibility that NASH is an actual feature of the metabolic syndrome [20].

So far, for NASH, a plethora of biomarkers and predictive models and scores have been proposed for the early diagnosis or the prediction of fibrosis, some of these even including single nucleotide polymorphisms of the *PNPLA3* gene [5]. The patatin-like phospholipase domain-containing protein 3 (PNPLA3), also known as adiponutrin, is an enzyme encoded by the *PNPLA3* gene. Variants of *PNPLA3* polymorphisms were proven to be associated with NAFLD/NASH, together with the polymorphisms of the transmembrane 6 superfamily member 2 (*TM6SF2*) gene [21]. A recent meta-analysis concluded that the *PNPLA3* rs738409 polymorphism was significantly associated with NAFLD, and G allele carriers more with frequently NASH [22]. It comes clear that NAFLD/NASH is associated with genetic variants which could explain other recent interesting findings that non-obese NASH patients have more advanced liver fibrosis than obese ones [23].

Paraoxonase-1 is an enzyme regarded nowadays as having important antioxidant and antiatherogenic properties due to the fact of three main activities (paraoxonase, arylesterase, and lactonase) influenced by the expression of the *PON1* gene [24]. In a previously published pilot study, we evaluated *PON1* gene polymorphisms in a small group of patients with NASH [25], and we found that the L55M polymorphism can be associated with NASH, while the PON1 activities were not significantly different in the NASH group versus a control group. Also, after the pilot study, we were able to estimate 73 as the minimum number of patients and controls to be included in a study with 80% statistical power at a significance level of 5%; we acted accordingly, including 81 subjects in each of the groups of the present study.

Our previous experience in the evaluation of PON1 biologic and genetic variants for different conditions [19,26,27] prompted us to evaluate the PON1 serum concentration (using modern, ELISA kits, with high sensitivity) as a parameter of its status instead of its three activities (with different results in the literature, even for the same diseases). Our idea was reinforced by a recent study by van den Berg et al. [28], who showed that serum PON1 activity can be paradoxically maintained in patients with NAFLD, despite low serum levels of HDL cholesterol. The different alloenzymes expressed by *PON1* gene single nucleotide polymorphisms (SNPs) express different concentrations and activities with proven implications in various diseases [29]. Thus, we chose for analysis the two important polymorphisms from the gene sequence (i.e., L55M and Q192R) and one from the promoter part of the gene, C-108T, which seems to be one of the main contributors to the regulation of PON1 clinical expression (at least 12% of PON1 variations can be attributed to this polymorphism) [10].

To our knowledge, this is the first study to evaluate PON1 serum concentration and the three main *PON1* gene polymorphisms in patients with NAFLD.

As expected, we found statistically significant differences between the NAFLD group and the control group regarding the clinical parameters. The NAFLD patients presented higher BMIs, waist circumferences, and more relatives with a history of cardiovascular diseases (Table 1). Also, they wore more prone to having pre-diabetes or diabetes, arterial hypertension or metabolic syndrome than the non-NAFLD group. Ischemic heart disease and angina pectoris were also significantly more frequent in the NAFLD cohort. These were expected results, because it is already known that NAFLD/NASH associates cardiometabolic comorbidities (mainly obesity, type 2 diabetes mellitus, and coronary artery disease) [30] and non-cardiometabolic comorbidities (e.g., chronic kidney disease, osteoporosis) [31].

We found the PON1 concentration to be statistically significantly reduced in the NAFLD group (*p* < 0.001) compared to the control group. In the multivariate linear regression, NAFLD was the only variable associated with PON1 concentration (*p* < 0.001). Regarding the genetic polymorphisms, we found only the presence of the M allele (LM + MM) of the L55M polymorphism to be associated with lower PON1 concentrations (*p* = 0.004). The heterozygous variant (LM) was present in 58% of the cases from the NAFLD group, while the mutated homozygous variant was present in 14.8% of the cases. The number of patients from our study was modest, so in a larger cohort we would probably see an increase in the MM genotype frequency. The frequency of the M allele was significantly higher in the NALFD group than in the control group. The trend shows clearly that the M allele is associated with the presence of NAFLD. The MM genotype frequency was higher in our study than other studies ranging from 1.5% [32] to 13.2% [33]. Both Q192R and C-108T had no statistically significant influence on PON1 concentrations (Table 4). The pathway from the genetic variants containing the M allele to an altered expression of PON1 concentration can be explained partially by the interference of non-genetic factors known to influence PON1 status such as diet, physical exercise, certain medications, etc. [34]. These are factors which are extremely difficult to evaluate during clinical studies. Also, we have previously reinforced that patients with NAFLD/NASH have a high plasmatic level of C-reactive protein (CRP), interleukin-6 (IL-6), and tumor necrosis factor-α (TNF-α), so a potential cytokines imbalance (not evaluated in the present study) could interfere also with the serum concentration of PON1 [35].

At this point, it is premature to emphasize that a low PON1 concentration might be implicated in NAFLD/NASH pathogenesis nor can we say that NAFLD might reduce PON1 concentration. It is well known that proinflammatory cytokines and an accumulation of reactive oxygen species (ROS) resulting from β- and omega-oxidation of free fatty acids (FFAs) lead to hepatic injury and progression from simple steatosis to NASH and advanced fibrosis [36]. Apart from its antioxidant effects, PON1 has multiple implications in metabolic disorders due to the fact of its promiscuous multiple activities. The enzyme was recently named a “regulator of glucose and lipid homeostasis” [37] which exerts its implication on the pathogenesis of NAFLD/NASH due to the fact of its protective function against oxidative stress and inflammation [37,38].

To date, studies that have evaluated PON1 activity in patients with NAFLD/NASH are scarce. A PubMed search using the keywords “Paraoxonase-1” and ”NAFLD” retrieved only 21 results. Moreover, many of these studies were performed on murine models (a limitation to “Humans” retrieved only 11 results), and one of them even advanced the hypothesis that PON1 could prove to be a promising marker for detection of NAFLD severity [39]. The only study that evaluated PON1 activity in a small NASH cohort (23 patients) was published by Baskol et al. [14] in 2005. In that study, serum PON1 activity was measured as the rate of paraoxon hydrolysis and was significantly decreased in the NASH group compared to the control group (*p* < 0.01). Two studies were published in 2019 which evaluated PON1 activity in patients with NAFLD. A study conducted by Fadaei et al. [40] showed decreased arylesterase and paraoxonase activities in patients with NAFLD (study conducted on 49 NAFLD patients), while the study performed by van den Berg et al. [28] showed no variation in PON1 arylesterase activity according to an elevated fatty liver index (FLI). Their findings and our findings may become a shifting point in future PON1 status evaluations in patients with NALFD/NASH, from serum activities to serum concentrations.

The present study has several limitations. First of all, NAFLD diagnosis was not established using a liver biopsy, as we have already explained. Even if we surpassed the estimation for the minimum number of patients required for statistical power, our study still had a reduced number of patients included. Also, serum levels of C-reactive protein, interleukin-6, and TNF-α were not evaluated in correlation with PON1 concentrations/genotypes, because their values were not available in all our patients.

Although paraoxonase-1 clinical research had its peak from 2010–2016, there are many studies since then, that have evaluated the status of PON1 in different pathologies [41,42,43]. Directions for further research should be related to simplifying PON1 serum measurements, practical inclusion of PON1 status (biologic and genetic) into scores for diagnosis and/or prognosis of diseases, and finding a potential PON1-enhancing drug to improve the antioxidant effect of PON1.

## 5. Conclusions

To our knowledge, this is the first study to evaluate PON1 concentrations and the *PON1* gene polymorphisms in patients with NAFLD. Patients from our study cohort had diminished PON1 serum concentrations compared to a non-NAFLD group. The LM + MM genotypes of the *PON1* gene L55M polymorphism were an independent predictor for the presence of NAFLD. In a multivariate linear regression model, NAFLD was linked to the PON1 serum concentration. These findings could open a path for further studies which could incorporate these parameters in either existent or new non-invasive scores, able to diagnose NAFLD and NASH without the need of a liver biopsy.

## Figures and Tables

**Table 1 jcm-08-02200-t001:** Characteristics of patients with and without non-alcoholic fatty liver disease (NAFLD).

Variables	Patients with NAFLD (*n* = 81)	Patients without NAFLD (*n* = 81)	*p*-Value
Age (years) *	51 (39; 60)	51 (39.5; 60.5)	0.96
BMI (kg/m^2^) *	30.69 (27.72; 33.70)	23.45 (21.81; 24.97)	<0.001
Waist circumference (centimeters) *	107 (94; 119)	79.8 (76; 87.75)	<0.001
Family history of cardiovascular diseases **	68 (84%)	13 (16%)	<0.001
Cigarette smoking **	19 (23.5%)	16 (19.8%)	0.78
Hypertension **	49 (60.5%)	32 (39.5%)	<0.001
Diabetes mellitus **	22 (27.2%)	3 (3.7%)	<0.001
Impaired fasting glucose and/or impaired glucose tolerance (pre-diabetes) **	23 (28.4%)	7 (8.6%)	0.001
Metabolic syndrome *	48 (59.3%)	10 (12.3%)	<0.001
Ischemic heart disease **	24 (29.6%)	10 (12.3%)	0.007
Stable angina pectoris **	16 (19.8%)	5 (6.2%)	0.010
Glycemia (mg/dL) *	109 (93; 129)	88 (81.5; 97.5)	<0.001
AST (UI/L) *	55 (47.5; 66)	23 (19; 30)	<0.001
ALT (UI/L) *	73 (62; 87.5)	23 (18.5; 29)	<0.001
ALP (U/L) *	213 (165; 267.5)	132 (111; 168)	<0.001
GGT (U/L) *	38 (30; 55)	22 (17; 30)	<0.001
Total cholesterol (mg/dL) *	211 (175.5; 235.5)	215 (178; 240)	0.35
HDL cholesterol (mg/dL) *	39 (35; 50)	52 (42; 60.5)	<0.001
Triglycerides (mg/dL) *	186 (115; 278)	120 (83.5; 151)	<0.001
Total bilirubin (mg/dL) *	0.75 (0.5; 1)	0.9 (0.6; 1.1)	0.13
PLT (10^3^/µL) *	222 (187; 259.5)	245 (202.5; 284.5)	0.01
Serum albumin (g/dL) *	4.47 (4.15; 4.98)	4.28 (4; 4.55)	<0.001

* Median value, (25th and 75th percentiles). ** Number of patients, (percentage). AST = aspartate aminotransferase, ALT = alanine aminotransferase, ALP = alkaline phosphatase, GGT = gamma-glutamyl transferase, HDL = High-density lipoprotein, PLT = platelets count.

**Table 2 jcm-08-02200-t002:** Paraoxonase 1 concentration and gene polymorphisms.

Variables		Patients with NAFLD (*n* = 81)	Patients without NAFLD (*n* = 81)	*p*-Value
Paraoxonase concentration (ng/mL) *		11.69 (10.88; 12.38)	15.10 (13.72; 16.16)	<0.001
Q192R polymorphism (rs662) **	QQ	38 (46.9%)	42 (52.9%)	0.807
	QR	37 (45.7%)	33 (40.7%)	
	RR	6 (7.4%)	6 (7.4%)	
	QR + RR	43 (53.1%)	39 (48.1%)	0.637
L55M polymorphism (rs854560) **	LL	22 (27.2%)	44 (54.3%)	0.002
	LM	47 (58.0%)	27 (33.3%)	
	MM	12 (14.8%)	10 (12.3%)	
	LM + MM	59 (72.8%)	37 (45.7%)	<0.001
C-108T polymorphism (rs705379) **	CC	31 (11.8%)	36 (11.8%)	0.707
	CT	39 (70.6%)	36 (58.8%)	
	TT	11 (17.6%)	9 (29.4%)	
	CT + TT	50 (61.7%)	45 (55.6%)	0.524

* Median (25th and 75th percentiles). ** Number of patients (percentage). QQ, LL, RR – wild homozygous variants; QR, LM, CT – heterozygous variants; RR, MM, TT – mutated homozygous variants.

**Table 3 jcm-08-02200-t003:** Multivariate binary logistic regression.

	B *	*p*-Value	(OR) **	95% CI for OR	
				Minimum	Maximum
BMI	0.747	<0.001	2.110	1.627	2.736
Family history of cardiovascular diseases	1.512	0.02	4.536	1.195	17.218
Genotypes LM + MM of the L55M polymorphism	1.235	0.04	3.439	1.048	11.285

* B: unstandardized beta. ** Odds ratio.

**Table 4 jcm-08-02200-t004:** Variables correlated to the paraoxonase-1 (PON1) concentration.

Variables		PON1 Concentration (ng/mL)	*p*-Value
Gender *	Female	12.85 (11.79; 15.08)	0.86
	Male	12.98 (11.26; 15.44)	
Smoking *	No	12.87 (11.68; 15.04)	0.77
	Yes	12.93 (11.39; 15.00)	
Family history of cardiovascular diseases *	No	14.21 (12.84; 15.68)	<0.001
	Yes	12.17 (11.23; 14.65)	
Hypertension *	No	13.90 (11.61; 15.71)	0.004
	Yes	12.12 (11.25; 13.65)	
Metabolic syndrome *	No	13.86 (11.69; 15.70)	<0.001
	Yes	12.09 (11.20; 13.41)	
Ischemic heart disease *	No	13.06 (11.59; 15.27)	0.16
	Yes	12.04 (11.10; 14.82)	
Stable angina pectoris *	No	13.00 (11.55; 15.30)	0.14
	Yes	12.03 (11.14; 14.14)	
Impaired fasting glucose and/or impaired glucose tolerance (pre-diabetes) *	No	13.17 (11.67; 15.39)	0.004
	Yes	11.81 (10.74; 13.49)	
Diabetes mellitus *	No	13.03 (11.55; 15.36)	0.11
	Yes	12.12 (11.26; 13.61)	
Q192R polymorphism *	QQ	12.98 (11.67; 15.48)	0.66
	QR	12.79 (11.30; 14.94)	
	RR	12.46 (11.03; 14.43)	
	QR + RR	12.59 (11.24; 14.79)	0.41
L55M polymorphism *	LL	13.77 (12.02; 15.94)	0.015
	LM	12.17 (11.21; 14.70)	
	MM	12.28 (11.20; 14.39)	
	LM + MM	12.22 (11.21; 14.50)	0.004
C-108T polymorphism *	CC	12.98 (11.61; 15.29)	0.8
	CT	12.97 (11.42; 15.02)	
	TT	12.30 (11.02; 14.98)	
	CT + TT	12.93 (11.26; 15.01)	0.57

* Indicated by Median concentration of PON1, (25th and 75th percentiles).

**Table 5 jcm-08-02200-t005:** Multivariate linear regression for PON1 concentration.

	Unstandardized Coefficients	t **	*p*-Value	95.0% CI for B	
	B *			Minimum	Maximum
Non-alcoholic fatty liver disease (NAFLD)	−0.120	−9.489	<0.001	−0.145	−0.095
Family history of cardiovascular disease	0.001	0.051	0.9	−0.023	0.024
Metabolic syndrome	0.022	1.766	0.07	−0.003	0.046
L55M polymorphism	−0.001	−0.138	0.8	−0.023	0.020

* B: unstandardized beta. ** t: t statistic.

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
