# Peer review of "Paraoxonase-1 Serum Concentration and PON1 Gene Polymorphisms: Relationship with Non-Alcoholic Fatty Liver Disease"

_jcm, 2019, doi:10.3390/jcm8122200_

Round 1

Reviewer 1 Report

In this study, Milaciu et al. seek the functional roles of PON1 polymorphisms in NASH. The authors use the cohort with 81 controls and 81 NASH patients. Results sound solid and I do not have major comments/criticisms for this manuscript. However, I feel like some descriptions are too strong and exaggerated. The title is good and appropriate, but the description like “NASH presence was the only variable linked with the low PON1 concentration” sounds too strong because numbers of factors are significantly different between two groups (Table 1) and several parameters influence PON1 concentrations (Table 4). Table 5 shows the association between NASH and PON1 concentrations in this cohort, but it does not necessarily prove that “only” NASH status induces low PON1 concentrations. Some descriptions are misleading and need to be elaborated. Figure 1 is also misleading and need to be edited, or I think Figure 1 is not necessary.

Author Response

Thank you for your comments!

Best regards,

Mircea Milaciu

Reviewer 2 Report

This original study is well-written, the employed methodology scientifically sound and the conclusions sufficiently backed by the provided data. Yet, some points of concern should be addressed before publication in the Journal of Clinical Medicine:

(1) The characterization of the “NASH cohort” is not complete and it cannot be correctly established if the patients suffer(ed) from NASH. Detection of hepatic steatosis and increased serum liver damage markers defines rather a more general “NAFLD condition” that might, or not, have concomitantly inflammation, fibrosis,… Therefore, it the use of NAFLD cohort vs a cohort without NAFLD seems more appropriate.

(2) Although a higher propensity of NAFLD is shown in patients carrying a specific L55M variant, it is also noticeable that only the heterogenic LM variant accounts for that difference. One would expect to observe this effect more stringently in the homozygous variant? This should be properly addressed in the discussion section. Could heterogenicity in the NAFLD cohort have played a role here?

(3) Table 4: “Variables influencing correlated to the PON1 concentration”

(4) Fig 1:
left arrow : predictor should be “indicator”;
lower rectangle: influencing should be “correlated to”
Right rectangle: these factors were not documented in this study!? Potential correlation with investigated parameters (documented in the tables of the study) should be mentioned here.

(5) Abstract: better define genotypes of L55M in the results paragraph. "LM+MM" as is currently stated is not comprehensible. Add abbreviation explanation and/or hetero-/homozygous to genotype variants... 

Author Response

Thank you for taking the time to evaluate our manuscript and for your advice! Please see the attachment.   Sincerely yours, Mircea Milaciu
